# Trends in adverse perinatal outcomes and associated hospitalisations, emergency department presentations, and healthcare costs from birth to early childhood in the Northern Territory, Australia: A two-decade population-based study

Tsegaye G. Haile[1,2]*, Gavin Pereira[1,3,4], Richard Norman[1], Gizachew A. Tessema[1,3,5]

**1** Curtin School of Population Health, Curtin University, Perth, Western Australia, Australia, **2** Department of Health Systems and Policy, Institute of Public Health, University of Gondar, Gondar, Ethiopia, **3** enAble Institute, Curtin University, Perth, Western Australia, Australia, **4** WHO Collaborating Centre for Climate Change and Health Impact Assessment, Faculty of Health Science, Curtin University, Perth, Western Australia, Australia, **5** School of Public Health, University of Adelaide, Adelaide, South Australia, Australia

\* t.haile2@postgrad.curtin.edu.au

## Abstract

Adverse perinatal outcomes, including preterm birth (PTB), small-for-gestational-age (SGA), and low birthweight (LBW), impact childhood health and impose substantial burdens. This retrospective cohort study included all births in the Northern Territory, Australia, from July 1, 2000, to June 30, 2016, examining trends in these outcomes and related hospitalisations, emergency department (ED) presentations, and healthcare costs through June 30, 2021. Births were linked to hospitalisation, ED, and cost-weight data. Cost, adjusted to June 2024 Australian Dollars (AUD), includes both direct medical and non-medical components. A Generalized Additive Model with a gamma distribution and log link was used to identify cost drivers. A total of 31,183 and 42,174 births were linked to hospitalisations and ED records, respectively. The incidence of PTB increased from 8.1% to 8.7%, while SGA declined from 15.2% to 11.3%. The mean number of hospitalisations by age five increased for children with PTB (1.3±0.7 to 6.9±6.0), and SGA (1.2±0.6 to 8.1±15.1), despite a decline in length of stay. ED presentations also increased for children with PTB (1.3±0.5 to 11.5±10.7), SGA (2.2±1.9 to 12.2±11.5), and LBW (1.2±0.2 to 10.9±8.7). Median five-year hospitalisations cost was AUD 23,848 (IQR: 11,858–44,475) for children with PTB and SGA, compared with AUD 8,668 (IQR: 4,365–17,855) for term non-SGA children. ED cost was AUD 3,108 (IQR: 1,609–7,520) versus AUD 2,058 (IQR: 1,032–4,057), respectively. Costs increased over time for SGA and LBW but declined slightly for PTB. Higher costs than the national average were observed among Indigenous children, those from remote areas, and those with prolonged hospital stays.

**Data availability statement:** All data are included in the manuscript and/or Supporting Information files.

**Funding:** This research was supported by the Australian National Health and Medical Research Council (NHMRC) Investigator Grant (#1195716) to GAT, and NHMRC Project Grant (#1099655) and Investigator Grant (#1173991) to GP. TGH is supported by a Research Training Program linked to the NHMRC. The funders had no role in study design, data collection and analysis, decision to publish, or preparation of the manuscript.

**Competing interests:** The authors have declared that no competing interests exist.

The healthcare burden associated with adverse perinatal has increased in recent cohorts, particularly among vulnerable groups. Future studies should quantify these burdens across population subgroups to better inform policy.

## Introduction

Despite global health advancements, perinatal morbidity such as preterm birth (PTB) [1,2], small-for-gestational-age (SGA), and low birthweight (LBW) [3], as well as stillbirth [4] and neonatal mortality, continue to impose a substantial dual burden of clinical and economic strain, with far-reaching short- and long-term consequences for individuals, healthcare systems, and society.

The prevalence, short- and long-term complications of perinatal morbidity and mortality, and health management, have been well documented in high-income countries such as Australia [5–7]. Accordingly, the trends and burden of perinatal morbidity and mortality have shown no significant decrease over time. As reported by the Global Burden of Disease 2021, PTB remains the leading cause of under-five mortality, followed by encephalopathy, congenital heart defects, and other disorders [5]. In 2021, the Australian Institute of Health and Welfare reported 9.6 perinatal deaths per 1,000 total births, 75% of which are stillbirths [8]. In the Northern Territory (NT), Australia, a region characterised greater relative socioeconomic disadvantage, this burden was more than double, with 20.6 perinatal deaths per 1,000 total births. Additionally, the region experiences elevated rates of PTB [5] and other perinatal morbidities, necessitating coordinated efforts to reduce the burden and meet national target levels. Due to the advancement of care and management the survival and outcomes of children born with adverse outcomes has been improved. Similarly, their length and frequency of health facility visit has increased [7,9,10]. These improvements might have profound economic burden to the healthcare system though more intensive resource use.

Only a small number of studies have measured the cost associated with PTB over limited time periods in Australia. In one example, a study by Callander et al. [11] examined the two-year cost of preterm and extremely preterm infants in Queensland, Australia, finding that the mean annual hospitalisation cost was $182,312 for extremely preterm babies (22–28 weeks gestation) in the first year and $9,958 in the second year. Another study by Stephens et al. [7] in New South Wales, Australia, assessed the hospitalisation and acute care costs of very and moderately preterm infants over the first six years of life, reported particularly high cost in the first year following birth. These studies focused solely on PTB and were limited to a few cohort years, covering both births and follow-up for a small number of years, making it difficult to capture how costs change with advancements in healthcare and efforts to meet national targets over time. Furthermore, they were limited to individual states, meaning we have a scarcity of evidence from the NT examining the health and economic burden of a border range of adverse perinatal outcomes [12]. The NT is one of the Australia's most remote and sparsely populated region with an estimated population of 232,605 according to the 2021 Population and Housing Census [13,14]. It has

a higher proportion of Indigenous residents compared to other Australian states, and higher proportion of health burden has been reported in the region.

Children born with adverse perinatal outcomes, such as PTB, SGA, and/or LBW, often require longer hospital stays, more frequent healthcare visits, and higher levels of intensive care. Consequently, they contribute to substantial health-care resource utilisation and impose both short- and long-term economic burdens on the health system. PTB, SGA, and LBW are distinct clinical outcomes, each with unique aetiologies. However, these outcomes may co-occur and share common risk factors, including maternal health conditions, socio-demographic disadvantages, and environmental exposures. These conditions are also associated with similar long-term health consequences, such as increased morbidity, developmental vulnerabilities, and mortality [15,16]. From a health systems perspective, PTB, SGA, and LBW represent a cluster of early-life adversities that place overlapping and sustained demands on healthcare services. To comprehensively assess the cumulative burden of adverse perinatal outcomes, a combined analytic approach that captures any instance of PTB, SGA, and/or LBW enables quantification of the total healthcare impact of these interrelated conditions. This approach is crucial for informing system-wide service planning and equitable resource allocation. Therefore, in this study we: (i) estimated the incidence and trends of adverse perinatal outcomes (PTB, SGA, and/or LBW); (ii) measured hospitalisation and emergency department (ED) presentation rates and trends; and (iii) analysed associated healthcare costs during the first five years of life over the past two decades, identifying the key drivers of these costs from a health systems perspective in the NT, Australia.

Understanding trends in the economic burden associated with adverse perinatal outcomes across early childhood, particularly among priority populations such as Indigenous and non–Indigenous children, and those residing in remote versus urban areas—can inform equitable resource allocation and guide the development of effective, cost-efficient perinatal health interventions. These findings also provide a foundation for estimating potential cost savings from initiatives aimed at reducing the incidence of adverse perinatal outcomes or improving perinatal health more broadly.

## Methods

### Study design and data linkage

We conducted a population-based retrospective cohort study using data on births from July 01, 2000, to June 30, 2016, in the NT, Australia. Administrative data were obtained by linking records from the Perinatal Trends dataset with the Inpatient and Emergency Department (ED) activity datasets. The Inpatient and ED activity datasets covered the period from July 1, 2000, to June 30, 2021. Additionally, cost-weight data were obtained from the National Hospital Cost Data Collection (NHCDC), specifically the Australian Refined Diagnosis Related Groups (AR-DRGs) for inpatient services, and Urgency Related Groups (URG) for ED costs, available from the Independent Health and Aged Care Pricing Authority [17].

The AR-DRGs provide a clinically meaningful way to relate or group the number and type of patients treated in an admission of care to the resources required in treatment and patients with similar AR-DRGs group are requiring similar hospital services. This classification has been updated every three years along with the International Statistical Classification of Diseases and Related Health Problems, Australian Classification of Health Interventions, Australian Coding Standards classification. Similarly, the URG classify patients presented to ED and group them based on their similar urgency services.

### Study population and cohort

Our study included all singleton live births from July 1, 2000, to June 30, 2016. Adverse perinatal outcomes in our study included PTB, SGA, and/or LBW. PTB, births occurring before 37 completed weeks of gestational age, is further classified as extreme PTB ($<27^{+6}$ weeks), very PTB ($28^{+0}$ to $31^{+6}$ weeks), moderate PTB ($32^{+0}$ to $33^{+6}$ weeks), and late PTB ($34^{+0}$ to $36^{+6}$ weeks) [18]. We classified the birthweight by gestational age for singleton births using the national birthweight percentiles by gestational age and classified them as SGA (<10th percentiles), appropriate-for-gestational-age (AGA) (10th

to 90[th] percentiles), and larger-for-gestational-age (LGA) (>90[th] percentiles) [19]. LBW was defined as a birthweight of less than 2,500 grams.

## Cost estimations and adjustment

In our study, we used an incidence-based approach of cost estimation which will highlight the likely savings from perinatal health programmes through reducing perinatal morbidity incidence or improvement of their health outcomes [20]. Accordingly, we examined the costs (cost of hospitalisation and ED presentation) related to PTB, LBW, and/or SGA. We estimated the cost from a health system perspective, including both direct and overhead costs. Direct costs refer to expenses directly related to health services, while overhead costs, which refer to indirect costs, include facility and utility services.

The cost weight per AR-DRG and URG from 2012 onwards are publicly available [17]. To estimate the cost weight per AR-DRG and URG for the period between 2000 and 2012, we deflated the 2012 value using the national medical services related consumer price index (CPI) [21]. Then we adjusted and reported all the cost to June 2024 value, in Australian Dollars (AUD), using the NT specific health related CPI. Finally, we estimated the incremental cost associated with adverse perinatal outcomes and the changes over time by calculating the difference in costs compared to those who presented to inpatient and ED services without adverse perinatal outcomes from birth to age five years.

## Inclusion criteria

Our study cohort included all singleton live births from July 1, 2000, to June 30, 2016. This timeframe ensured a complete five-year follow-up by June 30, 2021, enabling comprehensive assessment of hospitalisation and ED costs during the critical early childhood. Births with a birthweight below 400 grams or a gestational age less than 20 weeks were excluded, as they were considered non-viable. Children with congenital anomalies and those who died following birth hospitalisation were included in the cohort to ensure full representation of the population and its associated healthcare burden. Child age at admission was calculated using the admission date and birth date; as the perinatal dataset provided only month and year of birth, the first day of the month was assigned as the birth date (Fig 1).

## Statistical analysis

First, we conducted descriptive analyses, including frequencies and percentages, to provide an overview of the study population and outcome variables. We then estimated the incidence of adverse perinatal outcomes (PTB, SGA, and LBW) and the trends across years. Additionally, we calculated the rates of associated hospitalisations and/or ED presentations, reported as the mean or median number of hospitalisations and ED presentations per child-year. Finally, we described the mean or median cost per child and per admission or ED presentation from birth to age five years.

We used Generalised Additive Models (GAM) to identify the drivers of healthcare costs, allowing for non-linear associations with adverse perinatal outcomes [22,23]. Thin plate and cubic regression splines were applied to flexibly model smooth, non-linear relationships between costs and covariates. Model selection was based on comparison of the Akaike Information Criterion (AIC), residual maximum likelihood (REML), and R-squared values, with the gamma distribution and log link providing the best fit. Model diagnostics included examination of residual plots, smooth term uncertainty (via confidence intervals), and concurvity statistics to assess multicollinearity between smooth terms. We assessed overfitting through cross-validation and confirmed model stability through sensitivity analyses. Interaction effects were examined using tensor product smooths (ti() terms in the GAM framework) to capture non-linear interactions between key covariates, such as birthweight and gestational age.

Our adjustment variables were selected using a forward variable selection method. Covariates that were controlled for in GAM were selected based on published evidence including the clinical characteristics such as gestational age (weeks), birthweight (grams), Apgar score; evaluation of a newborn following birth using the criteria: Activity, Pulse, Grimace, Appearance, and Respiration, at one minute and five minutes, indications of congenital malformation of the baby that are

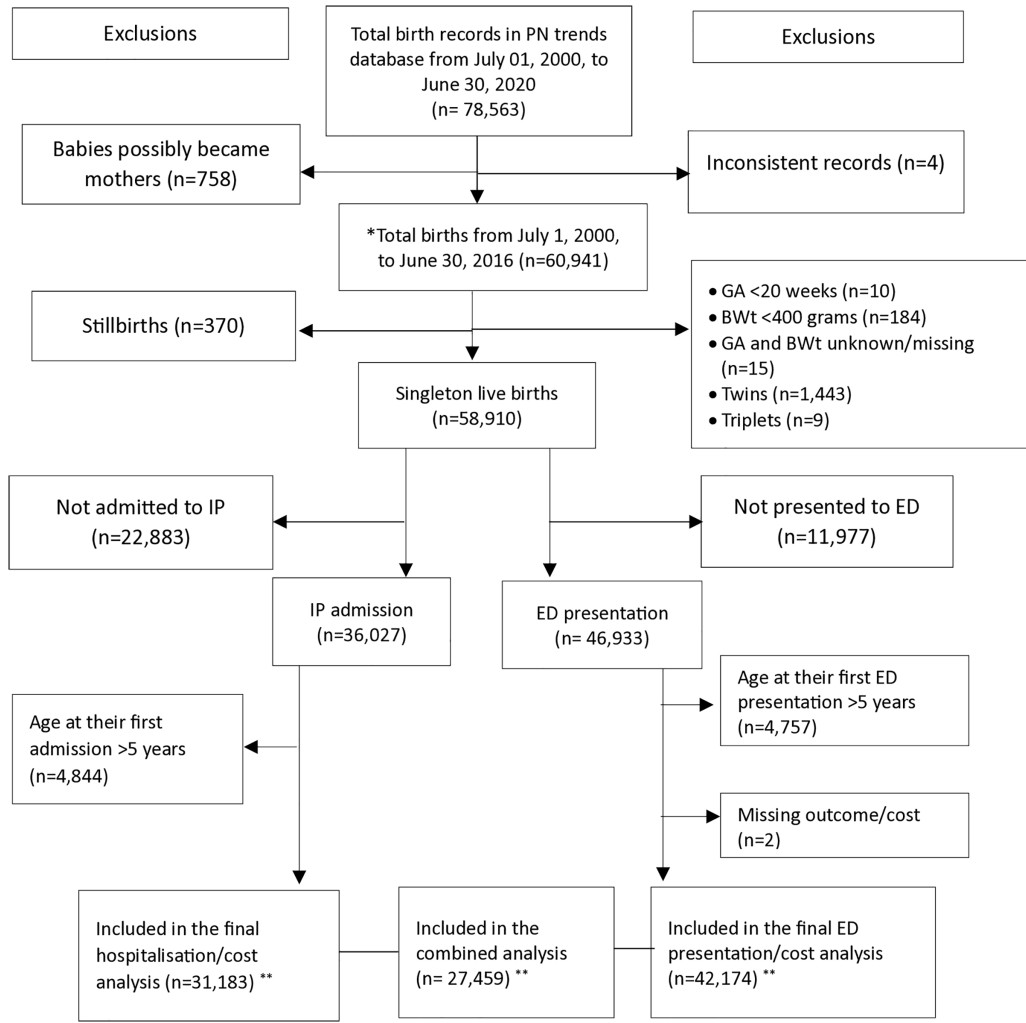

**Fig 1. Cohort selection flow diagram, NT, Australia, 2000–2016.** ED: Emergency Department, IP: Inpatient admission (also included birth hospital-isation), GA: Gestational age (in weeks), BWt: Birthweight (in grams),* These cohorts had a minimum of five years of follow-up as of June 30, 2021, and were selected to capture hospitalisations/ED presentations and the associated costs in the first five years of life.** These children included in the final analysis had at least one admission (for IP admission), at least one ED presentation (for ED presentation), or both (for combined) in the first five years of life.

present at birth (diagnosed, not diagnosed, under investigation, unknow), mode of separations of infant following birth hospitalisation (discharged to usual residence, transferred to an(other) acute hospital, died, left against advice or other), length of birth hospitalisation stay (days), length of hospital stay during the subsequent admission (days) (for hospitalisation cost), time spent at ED (hours) (for cost of ED presentation), and sociodemographic characteristics such as child sex at birth (male vs female), child (paternal) Indigenous status (Indigenous vs non-Indigenous), age of the mother at birth, remoteness of residence (remote vs urban, defined based on the Australian Statistical Geography Standard) and maternal marital status (single, widowed, divorced, separated, married/de facto), and other obstetric related characteristics such as parity, number of antenatal care visits during pregnancy, and whether it is a first pregnancy.

Finally, we fitted models for hospitalisation cost (model I), cost of ED presentation (model II), and combined cost for hospitalisation and ED presentation (model III), to identify the drivers of cost as follows.

Model I

$$g\left(E(\text{Hospitalisation cost})\right)$$
$$= \beta_0 + \beta_1(\text{Indigenous status of the mother}) + \beta_2(\text{Remoteness})$$
$$+ \beta_3(\text{Apgar score at 1 minute}) + \beta_4(\text{Apgar score at 5 minutes}) + \beta_5(\text{ANC visit})$$
$$+ \beta_6(\text{Congenital malformation status at birth}) + \beta_7(\text{First pregnancy})$$
$$+ \beta_8(\text{Mother's marital status}) + \beta_9(\textit{Mode of delivery})$$
$$+ \beta_{10}(\text{Child sex assigned at birth}) + \beta_{11}(\text{Birthweight - for - gestational - age at birth})$$
$$+ \beta_{12}(\text{Year of birth}) + \beta_{13}(\text{Country of birth})$$
$$+ \beta_{14}(\text{Mode of separations follwoing birth hospitalisation}) + \beta_{15}(\text{Parity})$$
$$+ S_1(\text{Age of the mother's}) + S_2(\text{Gestational age}) + S_3(\text{Birthweight})$$
$$+ S_4(\text{Length of stay following birth hospitalisation})$$
$$+ S_5(\text{Length of stay during readmission})$$

*Where $\beta_0$ is for intercept, $\beta_1$ to $\beta_{15}$ is for parametric coefficient (for categorical and linear variables), whereas $S_1$ to $S_5$ is for Estimated Degree of Freedom (EDF) for smoother terms (non-linear continuous variables).*

Model II

$$g\left(E(\text{ED cost})\right)$$
$$= \beta_0 + \beta_1(\text{Indigenous status of the mother }) + \beta_2(\text{Remoteness})$$
$$+ \beta_3(\text{Apgar score at 5 minutes}) + \beta_4(\text{ANC visit})$$
$$+ \beta_5(\text{Congenital malformation status at birth}) + \beta_6(\text{First pregnancy})$$
$$+ \beta_7(\text{Mother's marital status}) + \beta_8(\text{Child sex at birth})$$
$$+ \beta_9(\text{Mode of separation following birth hospitalisation}) + \beta_{10}(\text{Parity})$$
$$+ \beta_{11}(\text{Year of birth}) + S_1(\text{Age of the mother's}) + S_2(\text{Birthweight})$$
$$+ S_3(\text{Length of stay following birth hospitalisation}) + S_4(\text{Length of ED stay})$$

Model III

$$g\left(E(\text{Combined cost})\right)$$
$$= \beta_0 + \beta_1(\text{Indigenous status of the mother}) + \beta_2(\text{Birthweight}) + \beta_3(\text{Remoteness})$$
$$+ \beta_4(\text{Apgar score at 1 minute}) + \beta_5(\text{Apgar score at 5 minutes}) + \beta_6(\text{ANC visit})$$
$$+ \beta_7(\text{Congenital malformation status at birth}) + \beta_8(\text{First pregnancy}) + \beta_9(\text{Child age})$$
$$+ \beta_{10}(\text{Child sex at birth}) + \beta_{11}(\text{Mode of separation following birth hospitalisation})$$
$$+ \beta_{12}(\text{Parity}) + \beta_{13}(\text{Year of birth}) + \beta_{14}(\text{Age of the mother's})$$
$$+ \beta_{15}(\text{Mother's country of birth }) + S_1(\text{Length of stay following birth hospitalisation})$$
$$+ S_2(\text{Length of stay during readmission}) + S_3(\text{Length of ED stay})$$

**Missing data handling and sensitivity analysis**

We conducted a complete case analysis by excluding observations with missing values for the three main adverse perinatal outcomes, PTB, LBW, and SGA, as well as for variables included in the regression models. The frequency of missing data was low (<6 cases, representing <0.019%) and was unlikely to bias the results. To assess the robustness of our findings, we performed subgroup analyses based on the adverse perinatal outcomes: PTB, SGA, and/or LBW. Sensitivity analyses were also conducted to examine the impact of extreme values (outliers) for variables such as maternal age, gestational age, and the duration of hospitalisation and ED stays. For the ED time spent, we restricted the analysis to durations between 15 minutes and 24 hours, consistent with the National Emergency Access Target in Australia.

**Ethical approval and data access**

This study received ethical approval from the Human Research Ethics Committee of the Northern Territory Health and the Menzies School of Health Research (NT HREC; Ref #: 2022–4363) and from the Curtin University Human Research Ethics Office (Ref #: HRE2024–0039). Data were accessed on 02/02/2024. All data were de-identified, and the authors had no access to any information that could identify individual participants during or after data collection.

## Results

### Description of study participants

We included 58,910 singleton live births from July 1, 2000, to June 30, 2016. Of these, 31,183 births were linked to hospitalisation records, and 42,174 were linked to ED presentation records. These were included in the respective analyses of hospitalisation and ED presentation rates and its associated costs. Additionally, 27,459 births had records of both at least one hospital admission and one ED presentation; included in the combined costs of hospitalisations and ED presentations analyses.

Of these children admitted to inpatient care, 14.3% were PTB, 12.8% LBW, and 15.6% SGA, whereas among those children visited the ED before their fifth birthday, 13.5% were SGA at birth, 9.1% were born preterm, 7.9% were had LBW. The median length of hospital stays (during the subsequent admission) among these admitted was 3 (IQR: 1.4, 5.0) days (Table 1).

### Trends of adverse perinatal outcomes

We observed a slight increase in the incidence of PTB from 81 in 2000–87 per 1,000 live births in 2016. A similar trend was noted for LBW, which increase marginally from 77 in 2000–78 per 1,000 live births in 2016. In contrast, the incidence of SG) at birth decreased from 152 in 2000–113 per 1,000 live births in 2016 (Fig 2).

### Hospitalisations and ED presentations

We found that the overall mean number of hospitalisations per child was 3.1 (SD: 4.1) over the first five years. The mean number of hospitalisations per preterm child was 3.1 (SD: 3.5), per LBW child 3.2 (SD: 3.5), SGA was 2.8 (SD: 3.0), and per preterm child with LBW and SGA was 3.8 (SD: 3.4), while per term child with no LBW and SGA was 2.2 (SD: 2.3). Between 2000 and 2020, the mean hospitalisation increased for PTB (1.3 (SD:0.7) to 6.9 (SD: 6.0)), LBW births (1.3, SD: 0.6 to 7.2, SD: 5.2), and SGA births (1.2, SD: 0.6 to 8.1, SD: 15.1) in the first five years, with an exponential increase among children with SGA followed by LBW and PTB (Fig 3A). We found increasing hospitalisation rate over time, particularly 2017 onwards in all adverse perinatal outcomes. We found the overall median ED presentation per child was 3 (IQR: 1–6) over the first five years. Our findings also suggested that there is an increasing ED presentation across all adverse perinatal outcomes with a large change after the 2016. Children born preterm and with LBW presented to the ED more frequently than these children born SGA (Fig 3B). On the contrary, we observed a decreasing trend in hospital stays with each subsequent hospitalisation, while the length of stay after birth hospitalisation and ED presentations remained consistent across years for all adverse perinatal outcomes (S1 Fig).

### Cost of hospitalisations and ED presentations

We found that the overall median hospitalisations cost per preterm child with LBW and SGA was AUD 23,849 (IQR: 11,858–44,475), cost per child with at least one adverse perinatal outcome was AUD 12,373 (5,998–26,347), while the cost per term child with no SGA and LGA was AUD 8,668 (IQR: 4,365–17,855) in the first five years. Between 2000 and 2020, the median hospitalisation costs increased for SGA births (AUD 3,807, IQR: 3,527–6,175 to AUD 5,245, IQR: 3,016–9,931) and LBW births (AUD 5,234, IQR: 3,807–9,955 to AUD 5,843, IQR: 2,398–10959), while cost for PTB was

**Table 1. Description of the study participants, NT, Australia, 2000 –2020.**

| Variables | Hospitalisation (N = 31,183) | ED presentation (N = 42,174) | Combined (hospitalisation and ED presentation) (N = 27,459) |
|---|---|---|---|
| | n (%) | n (%) | n (%) |
| Gestational age | | | |
| EPTB (20$^{+0}$ – 27$^{+6}$ weeks) | 243 (0.8) | 127 (0.3) | 126 (0.5) |
| VPTB (28$^{+0}$ – 31$^{+6}$ weeks) | 528 (1.7) | 412 (1.0) | 410 (1.5) |
| MPTB (32$^{+0}$ – 33$^{+6}$ weeks) | 614 (2.0) | 498 (1.2) | 481 (1.7) |
| LPTB (34$^{+0}$ – 36$^{+6}$ weeks) | 3,060 (9.8) | 2,785 (6.6) | 2,496 (9.1) |
| Term (>=37$^{+0}$ weeks) | 26,738 (85.7) | 38,352 (90.9) | 23,946 (87.2) |
| Birthweight, mean (SD) (in grams) | 3,195.2(667.1) | 3,295.6(592.9) | 3,218.4(638.2) |
| LBW | | | |
| Yes | 3,999(12.8) | 3,344(7.9) | 3,120(11.4) |
| No | 27,184(87.2) | 38,830(92.1) | 24,339(88.6) |
| Sex assigned at birth | | | |
| Male | 17,171 (55.1) | 22,454(53.2) | 15,206(55.38) |
| Female | 14,005 (44.9) | 19,712(46.7) | 12,248(44.60) |
| Others | 7 (0.02) | 8(0.02) | 6 (0.02) |
| Birthweight-for-gestational-age | n = 31,176 | n = 42,166 | n = 27,454 |
| SGA | 4,870 (15.6) | 5,695(13.5) | 4,254(15.5) |
| AGA | 23,391 (75.0) | 32,537(77.2) | 20,670(75.3) |
| LGA | 2,915 (9.4) | 3,934(9.3) | 2,530(9.2) |
| Indigenous status (for baby) | | | |
| Indigenous | 17,392 (55.8) | 18,762(44.5) | 15,724 (57.2) |
| Non-Indigenous | 12,592 (40.4) | 20,389(48.3) | 10,646 (38.8) |
| Unknown | 1,199 (3.8) | 3,023(7.2) | 1,089 (4.0) |
| Remoteness | | | |
| Remote | 14,806 (47.5) | 16,213(38.4) | 13,177(48.0) |
| Urban | 16,374(52.5) | 25,946(61.5) | 14,280(52.0) |
| Unknown | ¥ | 15 (0.1) | ¥ |
| Birth hospitalisation outcome | n = 31,120 | n = 42,024 | n = 27,400 |
| Discharge to usual residence | 24,797 (79.7) | 36,773 (87.5) | 22,438 (81.3) |
| Transferred | 1,011 (3.2) | 868 (2.1) | 816 (3.0) |
| Died | 126 (0.4) | N/A | N/A |
| Refused care | 331 (1.1) | 345 (0.8) | 300 (1.1) |
| Others | 4,553 (14.6) | 3,715 (8.8) | 3,598 (13.1) |
| Unknow | 302 (1.0) | 323 (0.8) | 248 (0.9) |
| Indigenous status of the mother | | | |
| Indigenous | 17,392 (55.8) | 17,193(40.8) | 14,709(53.6) |
| Non-Indigenous | 12,592 (40.4) | 24,970(59.2) | 12,747(46.4) |
| Unknown | 1,199 (3.8) | 11(0.03) | ¥ |
| Age of the mother, mean (SD) (in years) | 26.6 (6.4) | 27.2(6.3) | 26.4(6.4) |
| Mother's country of birth | n = 31,172 | n = 42,157 | n = 27,449 |
| Australia | 27,620 (88.6) | 36,509 (86.6) | 24,502(89.3) |
| Others* | 3,468 (11.1) | 5,507 (13.1) | 2,887(10.5) |
| Unknown | 84 (0.3) | 141(0.3) | 60(0.2) |

*(Continued)*

**Table 1.** (Continued)

| Variables | Hospitalisation (N = 31,183) | ED presentation (N = 42,174) | Combined (hospitalisation and ED presentation) (N = 27,459) |
|---|---|---|---|
| | n (%) | n (%) | n (%) |
| First pregnancy | | | |
| Yes | 8,963(28.7) | 12,165(28.8) | 7,820(28.5) |
| No | 21,938(70.4) | 29,590(70.2) | 19,405(70.7) |
| Unknown | 282(0.9) | 419(1.0) | 234(0.8) |
| Parity | n = 31,176 | n = 42,163 | n = 27,452 |
| Null | 11,940(38.3) | 16,761(39.8) | 10,416(37.9) |
| One | 8,707(27.9) | 12,453(29.5) | 7,731(28.2) |
| Two to four | 9,325 (29.9) | 11,641(27.6) | 8,250(30.0) |
| Five and above | 1,204(3.9) | 1,308 (3.1) | 1,055(3.8) |

AGA: appropriate-for-gestational-age.

ED: Emergency Department.

EPTB: Extremely preterm birth.

LBW: Low birthweight.

LGA: Large-for-gestational-age.

LPTB: Late preterm birth.

MPTB: Moderate preterm birth.

SGA: Small-for-gestational-age.

VPTB: Very preterm birth.

*Others included Asia/China, New Zealand/Oceania, Americas, Africa/Middle East, Rest of Asia, UK/Ireland, and Rest of Europe.

¥ Values <6 are not reported as per the Human Research Ethics Committee (HREC) protocol.

N/A: No observation in the cell.

decreased, from AUD 6,581 (IQR: 3,807–11,368) to AUD 6,219 (IQR: 4,365–10,959). We found that the median hospitalisation cost per child born preterm within the first five years of life was AUD 19,362 (IQR: 9,056–35,339), while the cost per child born term was AUD 8,887 (IQR: 4,579–18,713) (S1 Table). The median hospitalisation cost per term child with AGA and normal birthweight was AUD 8,668 (IQR: 4,365–17,855), while the median cost per child with at least one morbidity was AUD 12,373 (IQR: 5,998–26,347). The mean cost for a child with all adverse perinatal outcomes (PTB, LBW, and SGA) was twice as high as the total mean cost of all births, while the mean cost for a child with at least one adverse perinatal outcome was 1.2 times higher (S2 Table). Based on the classification of preterm, the median hospitalisation cost per child per year for EPTB, VPTB, MPTB and LPTB were AUD 8,687 (IQR: 5,237–27,764), AUD 9,871 (IQR: 5,700–19,140), AUD 8,088 (IQR: 4,677–13,641), and AUD 6,500 (IQR: 4,311–9,620), respectively (S3-S5 Tables). The majority of the hospitalisation costs were incurred in the first year, with costs decreasing in subsequent years (Fig 4). However, we observed an increasing trend in hospitalisation cost over time till 2016 and then onwards a slight decreasing (Fig 5, S2 Fig and S3 Fig).

We found the median cost of ED presentation per preterm child with LBW and SGA in the first five years of age was AUD 3,108 (IQR: 1,609–7,520), while the cost per a term child without these outcomes was AUD 2,058 (IQR: 1,032–4,057). Between 2000 and 2020, the median cost of ED presentation for preterm child with SGA and LBW was increased from AUD 743 (IQR: 596–890) to AUD 815 (IQR: 599–990), compared with a term child with no SGA and LBW decreased from AUD 890 (IRQ: 823–890) to AUD 690 (IQR: 561–885) (S6 Table). The mean cost of ED presentation increased for SGA births (AUD 803 ± 147 to AUD 853 ± 355), while cost for LBW births (AUD 846 ± 107 to AUD 818 ± 322) and PTB (AUD 845 ± 108 to AUD 793 ± 295) was decreased (S7 Table). We observed an increase in the mean cost of ED presentations

**Fig 2. Trends of adverse perinatal outcomes, NT, Australia, 2000–2016. A**: Trends by adverse perinatal outcomes (SGA, PTB, and LBW), **B**: Trends by PTB categories (EPTB, VPTB, MPTB, LPTB and Overall PTB rate), **C**: Trends by birthweight-for-gestational-age percentile (SGA and LGA). EPTB: Extreme PTB (<28 weeks), LBW: Low birthweight, LGA: Large-for-gestational-age, LPTB: Late preterm birth ($35^{+0}$ to $36^{+6}$ weeks), MPTB: Moderate preterm birth ($33^{+0}$ to $34^{+6}$ weeks), PTB: Preterm births, SGA: Small-for-gestational-age, VPTB: Very preterm birth ($28^{+0}$ to $32^{+6}$ weeks).

per EPTB child per year over time (S8 Table). The mean cost of ED presentation has been decreasing across child age in all adverse perinatal outcomes (Fig 6).

Overall, the median cost (combined cost of hospitalisation and ED presentations) per a preterm child with SGA and LBW within the first five years of age was AUD 31,210 (IQR: 17,793); while the median cost per a term child without LBW and SGA was AUD 9,007 (IQR: 4,689–18,485). The cost has been decreasing across child age for all adverse outcomes (Fig 7).

## Drivers of cost of hospitalisations and ED presentations

We found that, after adjustment, hospitalisations cost was 34.8% (95% CI: 32.5 to 37.2) higher for children from Indigenous families compared to those children from non-Indigenous families. Hospitalisation costs were 8.2% (95% CI: 5.1 to 11.0) higher for children from rural areas compared to those from urban areas. A one-point increase in Apgar scores at 1 minute was associated with a 1.3% (95% CI: 0.3 to 2.3) reduction in hospitalisation cost. Children diagnosed and under investigation for congenital abnormalities incurred 44.7% (95% CI: 28.8 to 62.8) and 44.9% (95% CI: 32.8 to 58.2) higher cost of hospitalisation than other children. The birth not being the first for the mother was associated with higher hospitalisation costs by 4.1% (95% CI: 0.5 to 7.8) and a one child increase in parity was associated with a 2.5% (95% CI: 1.3 to 3.7) increase in hospitalisation cost. Emergency caesarean delivery was associated with a 4.3% (95% CI: 0.6 to 8.1)

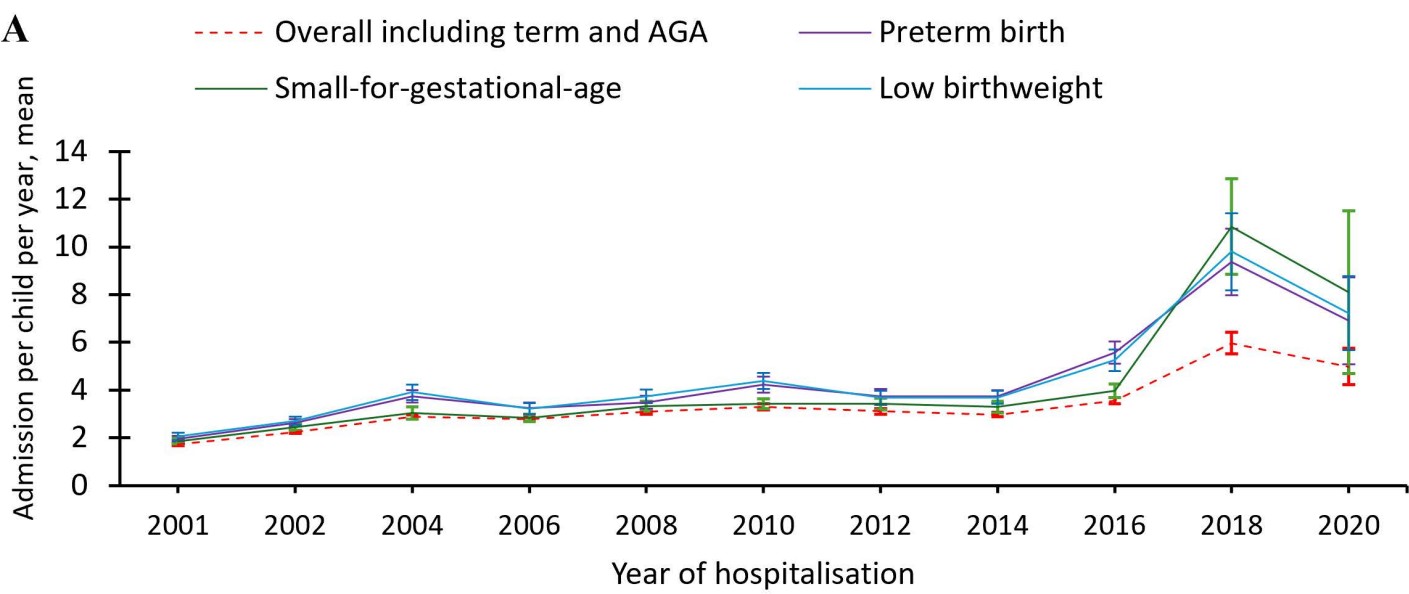

Global Public Health

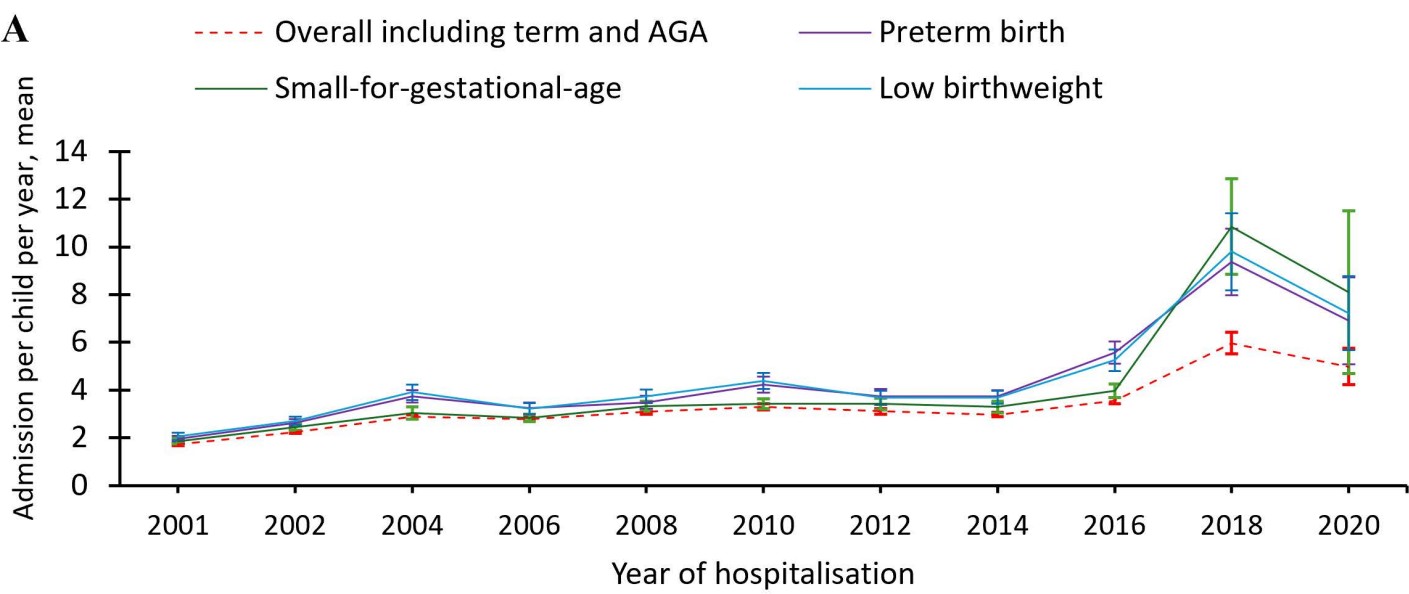

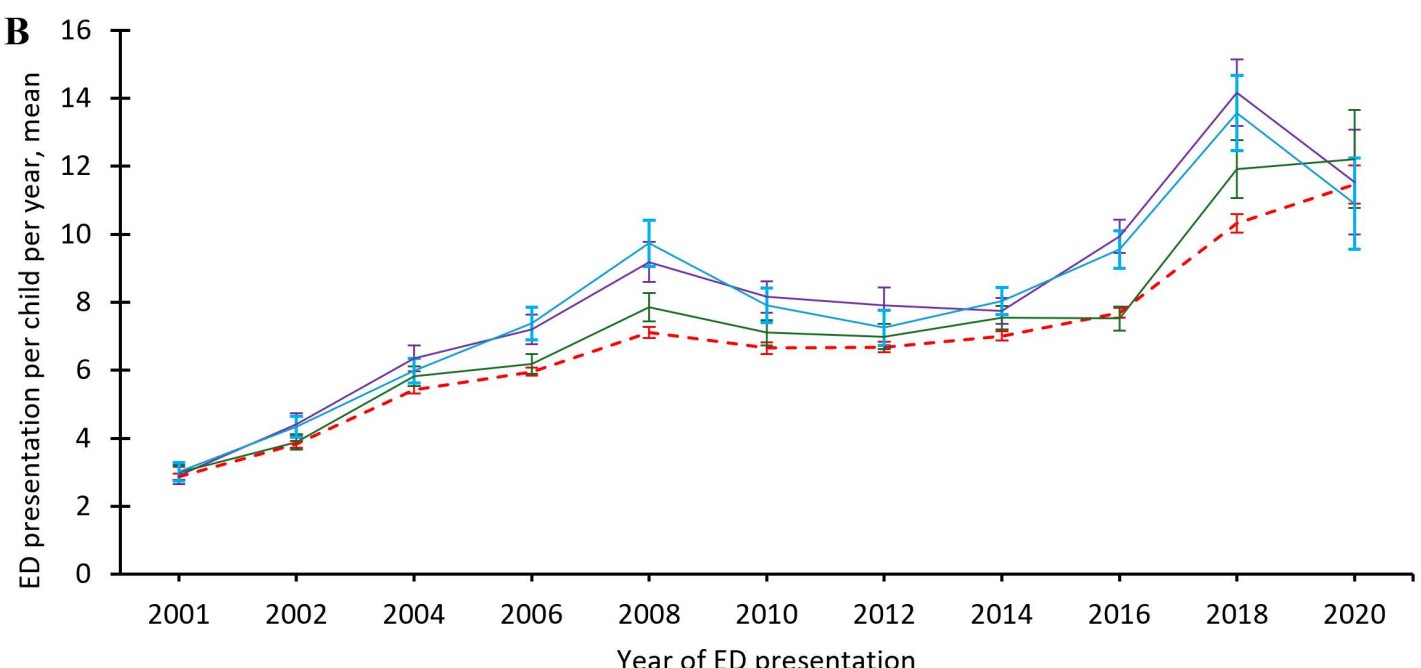

**Fig 3. Trends of hospitalisations and ED presentations by adverse perinatal outcomes, NT, Australia, 2000–2020.**

higher hospitalisation cost compared to spontaneous vaginal delivery. We observed a trend of increasing hospitalisation costs associated with adverse perinatal outcomes over time, with costs rising by up to 48.3% (95% CI: 33.1 to 65.4) in 2016 compared to the year 2000 (Table 2, S9 Table).

As shown in Figs 8A-E, we observed strong evidence for non-linear associations between hospitalisation cost and length of hospital stay, both following birth hospitalisation (EDF = 4.33) and during subsequent admissions (EDF = 8.89, as well as maternal age (EDF = 2.55) (S9-S10 Tables). Specifically, the association between length of hospital stay during

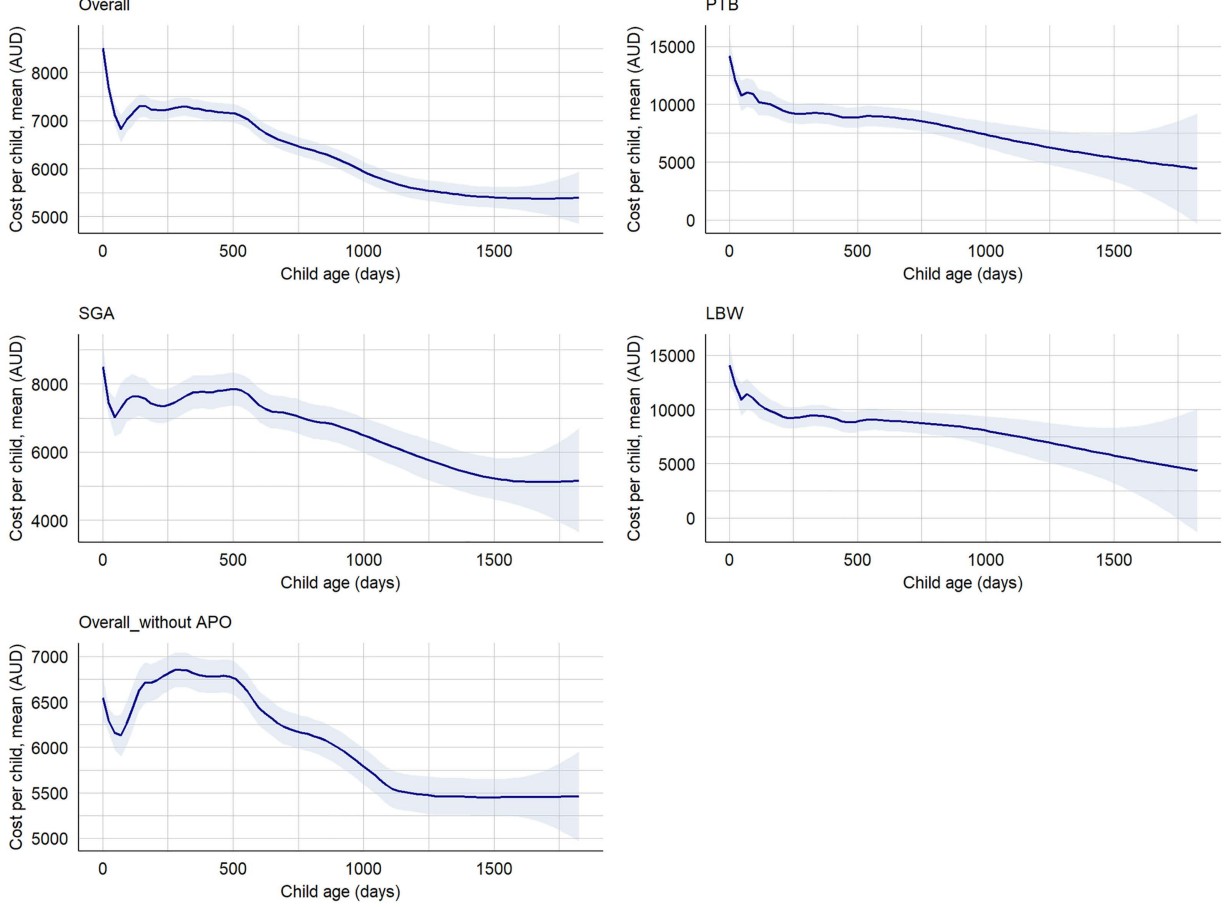

**Fig 4. Hospitalisations cost by child age and adverse perinatal outcomes, NT, Australia, 2000–2020.** APO: Adverse perinatal outcomes (PTB, LBW, and/or SGA), LBW: Low birthweight, PTB: Preterm births, SGA: Small-for-gestational-age, Overall_without APO: Defined as term births with appropriate-for-gestational-age.

subsequent admissions and cost was non-linear, with a peak was noted around two weeks days, followed by a slight decrease and another increase after approximately one month of age.

The association between birthweight and hospitalisation cost varied by gestational age. We observed that within certain ranges of birthweight (less than 3000 grams) and gestational age (less than 35 weeks), a decrease in birthweight was associated with a substantial increase in cost, particularly at lower gestational ages (S4 Fig). Correspondingly, the variability in the non-linear relationship between birthweight and gestational age increased significantly at the lower extremes of both birthweight and gestational age.

We found that cost of ED presentation was 9.3% (95% CI: 8.3 to 10.3) higher for children from Indigenous families compared to those children from non-Indigenous families. The cost was 5.1% (95% CI: 4.2 to 6.0) lower for children from urban areas compared to those from rural areas. A one-point increase in Apgar scores at 5 minutes was associated with a 1.3% (95% CI: 0.7 to 1.9) reduction in ED presentation costs. Children diagnosed and under investigation for congenital abnormalities incurred 4.8% (95% CI: 0.6 to 9.2) and 5.7% (95% CI: 2.6 to 8.9) cost of ED presentation than children who were not diagnosed congenital anomalies, respectively. We observed a trend of increasing cost of ED presentation associated with adverse perinatal outcomes over time, with costs rising by up to 18.0% (95% CI:13.8 to 22.1) in 2016 compared to the year 2000 (Table 3, S11 Table).

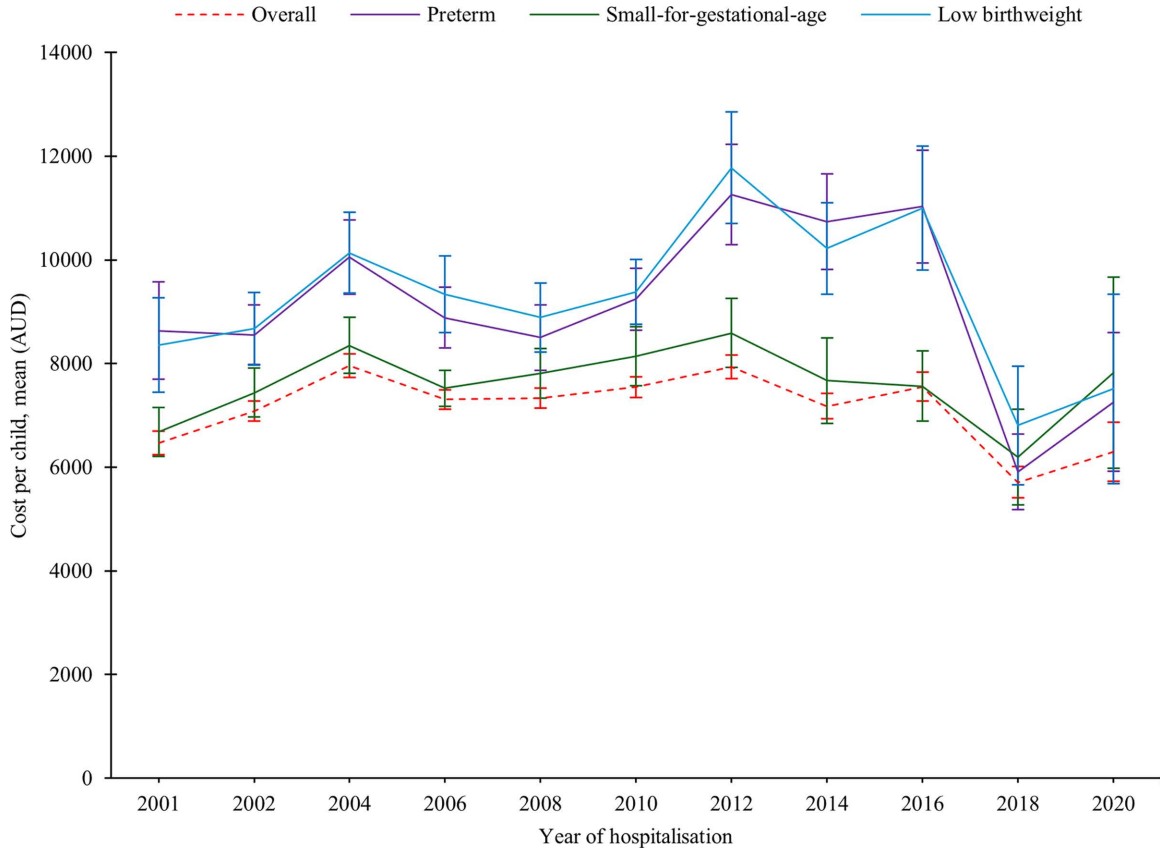

**Fig 5. Trends of hospitalisations cost by adverse perinatal outcomes, NT, Australia, 2000–2020.**

As shown in Figs 9A-E, we observed significant non-linear associations between cost of ED presentation and length of hospital stay, both following birth hospitalisation (EDF = 2.62) and ED presentation (EDF = 4.31), birthweight (EDF = 4.93) as well as maternal age (EDF = 3.82) (S11-S12 Tables). The association between birthweight and cost of ED presentation varied by gestational age. We observed that in the lower ranges of birthweight and gestational age, a substantial increase in cost of ED presentation (S5 Fig). In the combined cost drivers analysis, we identified a range of covariates that affect the cost (S13 Table).

## Discussion

Our study is the first to comprehensively assess long-term trends in adverse perinatal outcomes, such as PTB, SGA, and LBW, alongside their associated hospitalisation, ED presentation rates and healthcare costs from birth to age five years in NT, Australia. We identified adverse perinatal outcome rates (per 1,000 live births) of 87 for PTB, 128 for SGA infants, and 77 for LBW infants, contributing to rising healthcare costs over the last two decades. Although healthcare costs per child are comparable to those in other Australian States and Territories, the relative burden on the NT healthcare system is higher. The total proportional cost of morbidity (a child with at least one adverse perinatal outcome) is 45% greater when compared to children without morbidity (term children without LBW and SGA), highlighting the economic challenges of these adverse perinatal outcomes to the health system in this region. By the age of five years, the median healthcare cost for a preterm child with SGA and LBW was AUD 31,210, which included AUD 23,848 for hospitalisations and AUD 3,108 for ED presentations, compared with a term child without LBW and SGA was AUD 9,007 including hospitalisation cost of

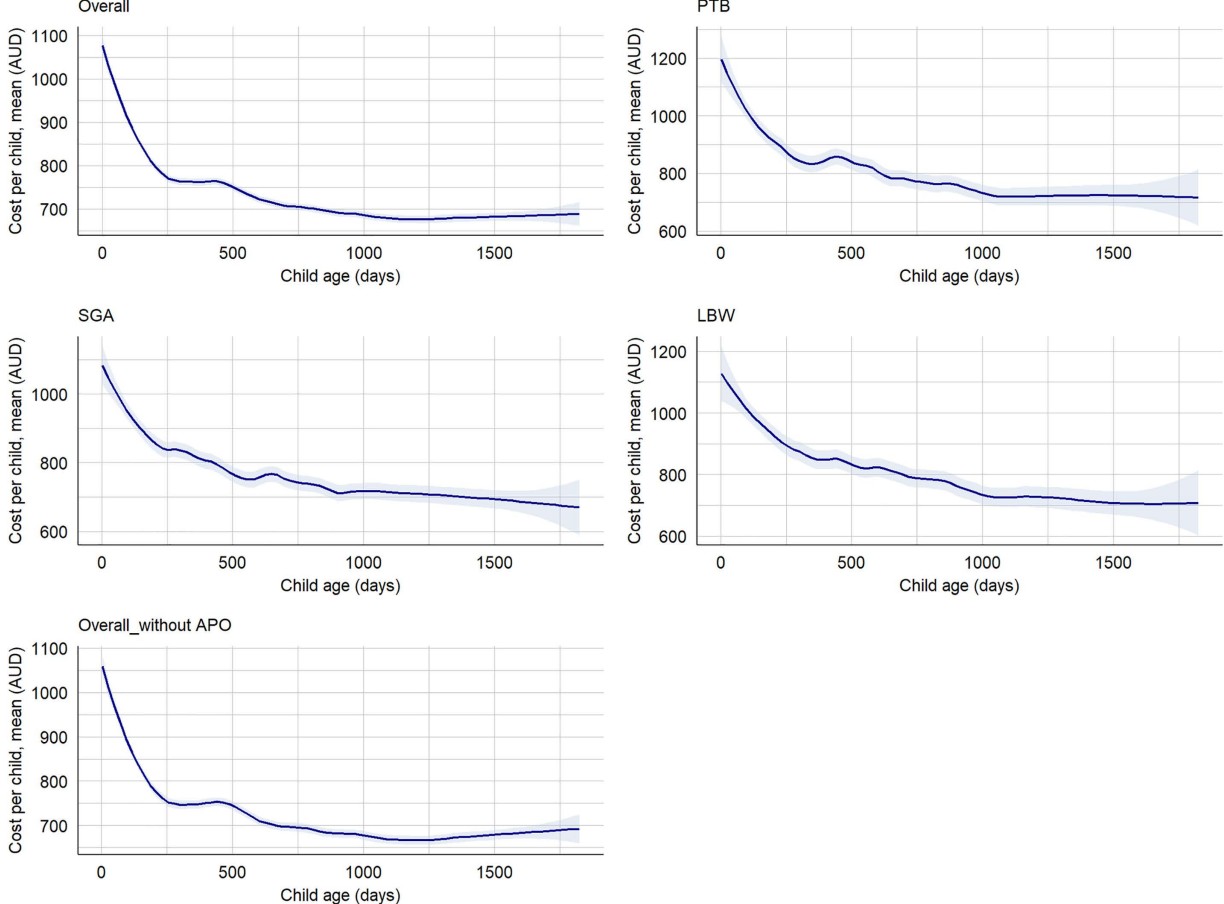

**Fig 6. Cost of ED presentations by child age and adverse perinatal outcomes, NT, Australia, 2000–2020.** APO: Adverse perinatal outcomes (PTB, LBW, and/or SGA), LBW: Low birthweight, PTB: Preterm births, SGA: Small-for-gestational-age, Overall_without APO: Defined as term births with appropriate-for-gestational-age.

AUD 8,668 and ED cost of AUD 2,058. Between 2000 and 2020, median hospitalisation costs rose for SGA (AUD 3,807 to AUD 5,245) and LBW births (AUD 5,234 to AUD 5,843) but declined for PTB births (AUD 6,581 to AUD 6,219). Median ED costs increased for preterm children with SGA and LBW (AUD 743 to AUD 815) but decreased for term children without these outcomes (AUD 890 to AUD 690). These results highlight the dual health and financial burden associated with adverse perinatal outcomes during early childhood. Addressing these challenges through preventative strategies and early interventions is essential to reduce healthcare costs and improve long-term health outcomes [24–27].

Our findings show an increasing trend in PTB and LBW, while a slight decrease in SGA was observed over time. This is consistent with previous studies, which reported a slight increasing trends in PTB, particularly among Indigenous populations [5], as well as a study by Yeung et al. [28] in Canada. Similarly, we observed an increase in hospitalisations and ED presentations over the past decades, particularly from 2016 onwards. Our findings are supported by other studies from other Australian states [7] and the US, which also reported higher hospitalisation rates among children with adverse perinatal outcomes such as PTB. The higher rates of hospitalisation and ED presentation observed in our study may partly be attributed to the mumps outbreak in the regional Australia, particularly in the Northern and Western Australia, from 2015 to 2017 [29,30], which increased hospital visits. In contrast, our findings show a decrease in the length of stay per readmission over the past decades for all adverse perinatal outcomes, aligning with a study in Canada [31]. However,

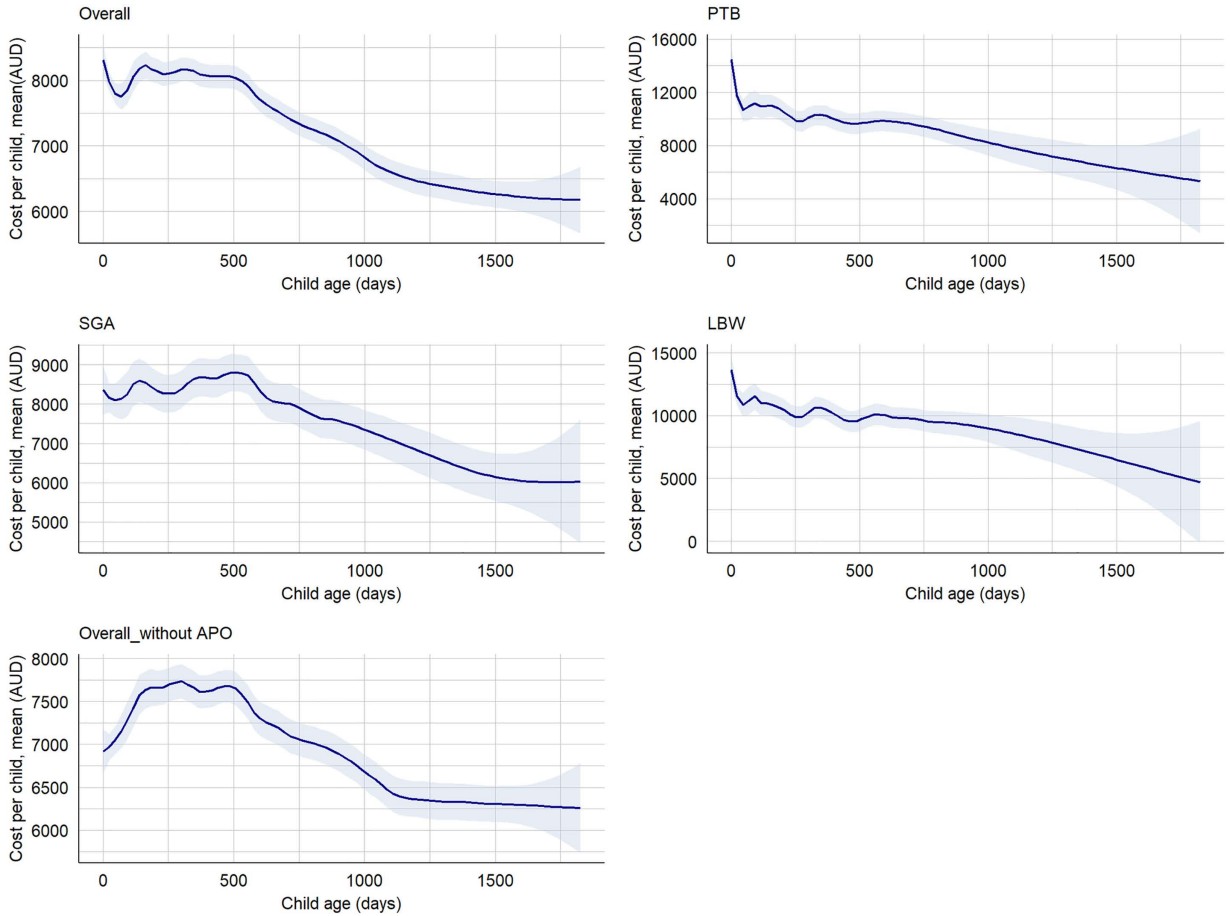

**Fig 7. Cost of hospitalisations and ED presentations by child age and adverse perinatal outcomes from birth to age five years, NT, Australia, 2000–2020.** APO: Adverse perinatal outcomes (PTB, LBW, and/or SGA), LBW: Low birthweight, PTB: Preterm births, SGA: Small-for-gestational-age, Overall_without APO: Defined as term births with appropriate-for-gestational-age.

a contrasting study from the same country reported an increase in the length of stay for extreme PTB (<24 weeks) [28]. Other research found no association between length of hospital stay and the risk of readmission [32,33]. We observed a consistent pattern in the length of hospital stay following birth hospitalisation and ED presentations. The median length of stays following birth hospitalisation was longer than subsequent admissions, particularly for children born LBW and PTB, compared to term children and those with appropriate birthweight. Previous studies have similarly shown that children born with adverse birth outcomes tend have longer hospital stays before discharge [7]. Conversely, the length of birth hospitalisation, along with subsequent admissions and ED presentations, required greater levels of intervention and were more intensive in the period immediately surrounding the time of birth. Our findings also suggest that longer hospital stays increase hospitalisation costs.

Certain clinical characteristics of the child are associated with increased healthcare costs. We observed a non-linear relationship between costs and factors such as gestational age, birthweight, length of hospital stays (including both birth hospitalisation and subsequent admission or ED stays), and child age. Birth at extreme gestational ages and birthweight were linked with increased financial cost for the healthcare system. Specifically, the impact of LBW on hospitalisation cost is more pronounced in preterm infants, highlighting the importance of targeted interventions for preventing PTB and LBW such as corticosteroids and progesterone administration for high-risk pregnancies, as well as nutritional support during

**Table 2. Drivers of hospitalisation cost from birth to age five years, NT, Australia, 2000–2020.**

| Variables | % Change | Lower 95% CI | Upper 95% CI |
|---|---|---|---|
| Indigenous status of mother | | | |
| Indigenous | Ref. | | |
| Non-Indigenous | -35 | -37.2 | -32.3*** |
| Remoteness of residence | | | |
| Rural | Ref. | | |
| Urban | -8.2 | -11.0 | -5.1*** |
| Apgar score at 1 minute | -1.3 | -2.3 | -0.3* |
| Apgar score at 5 minutes | -1.2 | -3.0 | 0.6 |
| Frequency of antenatal care visits | -0.1 | -0.3 | 0.3 |
| Congenital malformation status at birth | | | |
| Not diagnosed | Ref. | | |
| Diagnosed | 44.7 | 28.8 | 62.8*** |
| Under investigation | 44.9 | 32.8 | 58.2*** |
| Unknown | 6.7 | -0.1 | 13.9 |
| First pregnancy | | | |
| Yes | Ref. | | |
| No | 4.1 | 0.5 | 7.8** |
| Mother's marital status | | | |
| Single | Ref. | | |
| Married | 12.2 | 8.9 | 15.4*** |
| Others* | 25.2 | 18.4 | 32.5*** |
| Birth hospitalisation outcome | | | |
| Discharge to usual residence | Ref. | | |
| Transferred to acute care facility | 54.3 | 42.1 | 67.6*** |
| Refused care | 16.0 | 2.7 | 30.9* |
| Others | 9.3 | 5.3 | 13.5*** |
| Unknown | 24.0 | 5.7 | 45.3** |
| Sex at birth | | | |
| Male | Ref. | | |
| Female | -8.0 | -10.3 | -5.6*** |
| Parity* | 2.5 | 1.3 | 3.7*** |
| Mode of delivery | | | |
| SVD | Ref. | | |
| Breech | 8.2 | -15.0 | 37.7 |
| Forceps | 1.1 | -7.0 | 9.8 |
| Vacuum | -2.4 | -8.0 | 3.6 |
| CS elective | -1.9 | -6.2 | 2.4 |
| CS emergency | 4.3 | 0.6 | 8.1* |
| Birthweight-for-gestational-age | | | |
| SGA | 3.8 | -1.5 | 9.2 |
| AGA | Ref. | | |
| LGA | 1.0 | -5.4 | 7.6 |
| Year of birth | | | |
| 2000 | Ref. | | |
| 2001 | 5.0 | -4.0 | 15.0 |

*(Continued)*

**Table 2.** (Continued)

| Variables | % Change | Lower 95% CI | Upper 95% CI |
|---|---|---|---|
| 2002 | 11.5 | 1.7 | 22.3* |
| 2003 | 16.3 | 5.6 | 28.1** |
| 2004 | 27.1 | 15.4 | 40.0*** |
| 2005 | 26.5 | 15.0 | 39.2*** |
| 2006 | 28.8 | 17.1 | 41.7*** |
| 2007 | 33.0 | 20.8 | 46.3*** |
| 2008 | 27.5 | 16.0 | 40.2*** |
| 2009 | 38.8 | 26.2 | 52.6*** |
| 2010 | 33.1 | 21.1 | 46.3*** |
| 2011 | 35.2 | 23.0 | 48.7*** |
| 2012 | 39.6 | 27.0 | 53.5*** |
| 2013 | 41.9 | 29.1 | 56.0*** |
| 2014 | 42.2 | 29.5 | 56.3*** |
| 2015 | 49.6 | 36.2 | 64.5*** |
| 2016 | 48.3 | 33.1 | 65.4*** |
| Mother's country of birth | | | |
| Australia | Ref. | | |
| Others** | -2.0 | -5.3 | 3.4 |

Significant codes: ***<0.001, **<0.01, *<0.05.

AGA: Appropriate-for-gestational-age.

CS: Caesarean Section.

LGA: Large-for-gestational-age.

SGA: Small-for-gestational-age.

SVD: Spontaneous vaginal delivery.

* Others included separated, divorced or widowed.

** Others included region of birth categories: Asia/China, New Zealand/Oceania, Americas, Africa/Middle East, Rest of Asia, UK/Ireland, and Rest of Europe.

*Parity: Number of previous live or stillbirths (ranged from 0 to 13).

pregnancy [34–37], to prevent PTB and LBW. Healthcare costs initially decrease, then peak around two weeks of age before gradually decreasing again as the child aged. The observed non-linear relationship between cost and child age suggests that while post-birth hospitalisation discharge costs may decrease, infants remain vulnerable to infections, complications related to congenital conditions, and issues with feeding or growth, all of which contribute to ongoing healthcare needs. Previous studies support our findings that costs are notably higher for neonates born extremely and very preterm compared to those born late preterm, as they require more support during their initial hospitalisation [38]. Furthermore, infants born extremely (20–27+6 weeks) and very preterm (28+0 to 32+6 weeks) are more likely to develop comorbidities than those children born at full term, both in the short-term and later in life [39–41]. Infants born extremely preterm require frequent follow-up visits as they age, leading to continued healthcare costs over time [11,42]. We found that more than a third of the total cost related to adverse perinatal outcome has been incurred in the first few months following births, and then as their age increases, the direct and indirect cost to the health system reduces. This finding is supported by a study by Jacob et al. (2017), [43] which examined the cost effects of PTB, and showed that the cost decreases as the child age increases. Other studies have also identified that children's clinical characteristics affect the cost of hospitalisations among children with adverse outcomes [44,45]. This might be explained by the fact that infants born preterm and

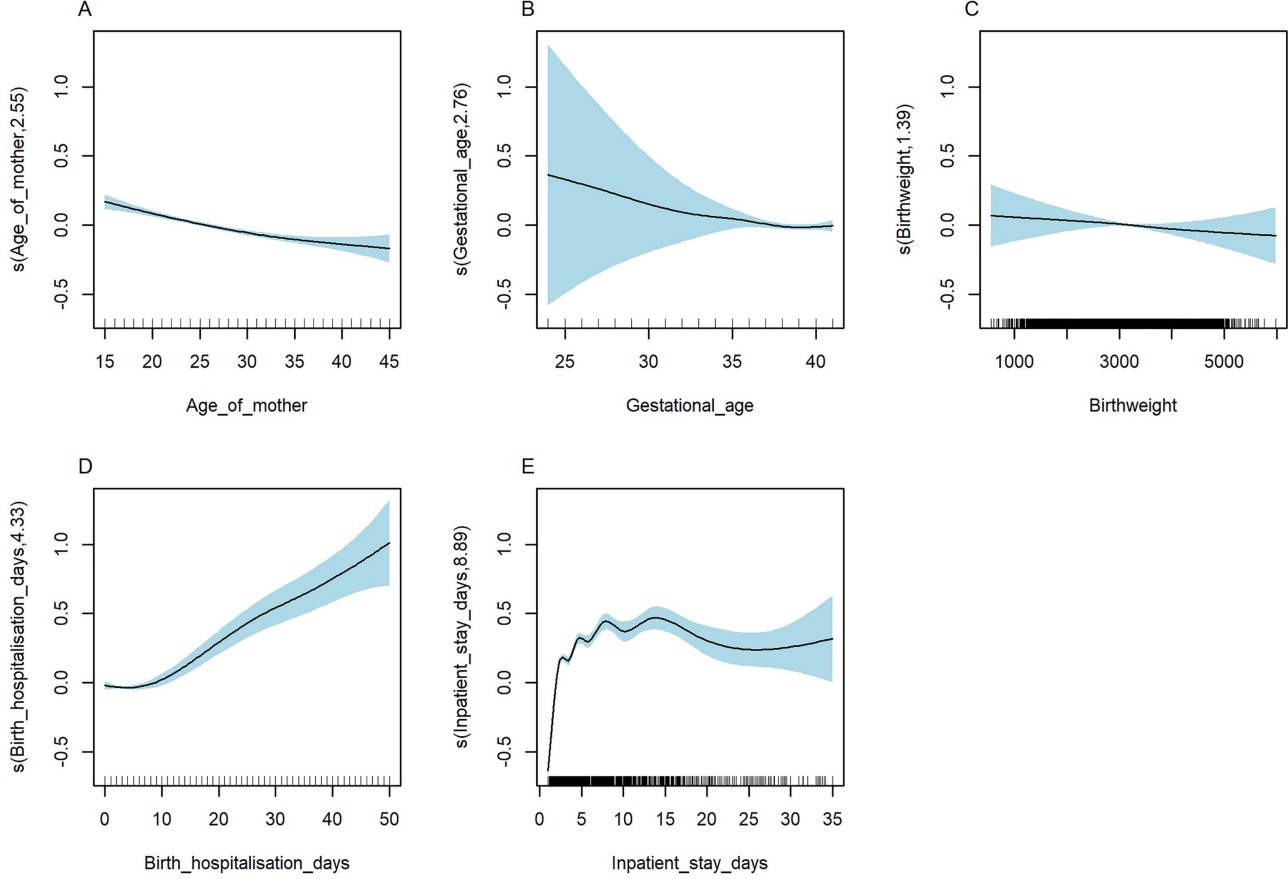

**Fig 8. Non-linear relationship between hospitalisation costs and covariates, NT, Australia, 2000–2020.** s(variable, df): Smooth function applied to the covariates, where df denotes the degree of freedom or smoothing parameter, indicating the flexibility of the non-linear relationship between the covariates and hospitalisation costs.

with LBW often require longer hospital stays after birth to gain the necessary weight and maturity before discharge, which places an additional burden on healthcare resources [46].

Vulnerable populations, particularly children of Indigenous parents and those residing in remote areas, incur higher healthcare costs compared to the average, which may place a disproportionate burden on the healthcare system. Our findings are consistent with another study in Australia, which highlight the higher healthcare utilisation and costs among vulnerable children during their early childhood years [47].

The strengths of our study include the use of a two-decade, population-based data, which allows for estimation of long-term trends in adverse perinatal outcomes and their associated hospitalisation and ED presentations and financial burden on the healthcare system. We included a broader range of adverse outcomes, encompassing major perinatal morbidity, and examined costs from birth up to age five, as this period captures most childhood morbidity following adverse outcomes [48,49] and reflects the long-term burden across decades.

Our study has some notable limitations. We deflated the costs back to 2000 using the 2012 CPI as the base year. However, this approach may not account for short-term fluctuations in healthcare costs resulting from economic disruptions or policy changes, nor does it fully capture advancements in care efficiency or the increasing use of costly medical innovations over time. As the NT Perinatal Trends dataset share only the month and year of birth, child age was estimated using

**Table 3. Drivers of cost of ED presentations from birth to age five years, NT, Australia, 2000–2020.**

| Variables | % Change | Lower 95% CI | Upper 95% CI |
|---|---|---|---|
| Indigenous status of baby | | | |
| Indigenous | Ref. | | |
| Non-Indigenous | -9.3 | -10.3 | -8.3*** |
| Remoteness of residence | | | |
| Rural | Ref. | | |
| Urban | -5.1 | -6.0 | -4.2*** |
| Apgar score at 1 minute | 0.3 | -0.1 | 0.6 |
| Apgar score at 5 minutes | -1.3 | -1.9 | -0.7*** |
| Antenatal care visits | -0.1 | -0.2 | 0.0** |
| Congenital malformation status at birth | | | |
| Not diagnosed | Ref. | | |
| Diagnosed | 4.8 | 0.6 | 9.2* |
| Under investigation | 5.7 | 2.6 | 8.9*** |
| Unknown | 3.9 | 1.7 | 6.2*** |
| First pregnancy | | | |
| Yes | Ref. | | |
| No | 2.3 | 1.2 | 3.3*** |
| Mother's marital status | | | |
| Single | Ref. | | |
| Married | 1.4 | 0.4 | 2.3** |
| Others | 1.7 | 0.1 | 3.3* |
| Unknown | 2.8 | -2.0 | 7.9 |
| Birth hospitalisation outcome | | | |
| Discharge to usual residence | Ref. | | |
| Transferred to an(other) acute care facility | -0.1 | -3.0 | 2.9 |
| Refused care | 9.8 | 5.4 | 14.4*** |
| Other | 6.1 | 4.6 | 7.5*** |
| Unknown | 5.4 | -0.1 | 11.2 |
| Sex at birth | | | |
| Male | Ref. | | |
| Female | -1.7 | -2.5 | -0.9*** |
| Parity | 1.5 | 1.1 | 1.9*** |
| Mode of delivery | | | |
| SVD | Ref. | | |
| Forceps | -2.0 | -4.4 | 0.4 |
| Vacuum | 0.1 | -1.7 | 1.8 |
| CS elective | -0.8 | -2.1 | 0.4 |
| CS emergency | -1.2 | -2.3 | 0.0 |
| Birthweight-for-gestational-age | | | |
| SGA | -0.9 | -2.7 | 0.9 |
| AGA | Ref. | | |
| LGA | 0.4 | -1.7 | 2.6 |
| Year of birth | | | |
| 2000 | Ref. | | |
| 2001 | -4.9 | -7.8 | -2.0** |

*(Continued)*

**Table 3.** (Continued)

| Variables | % Change | Lower 95% CI | Upper 95% CI |
|---|---|---|---|
| 2002 | -6.9 | -9.8 | -3.9*** |
| 2003 | -6.2 | -9.1 | -3.1*** |
| 2004 | -3.9 | -6.9 | -0.7* |
| 2005 | -3.0 | -6.1 | 0.1 |
| 2006 | -3.1 | -6.1 | 0.1 |
| 2007 | -3.5 | -6.6 | -0.4* |
| 2008 | -2.5 | -5.5 | 0.6 |
| 2009 | -1.6 | -4.6 | 1.6 |
| 2010 | 0.7 | -2.4 | 3.9 |
| 2011 | 0.6 | -2.4 | 3.8 |
| 2012 | 3.0 | 0.0 | 6.4 |
| 2013 | 10.0 | 6.6 | 13.5*** |
| 2014 | 13.6 | 10.1 | 17.2*** |
| 2015 | 16.0 | 12.4 | 19.6*** |
| 2016 | 18.0 | 13.8 | 22.1*** |

Significant codes: ***<0.001, **<0.01, *<0.05.

AGA: Appropriate-for-gestational-age.

CS: Caesarean Section.

ED: Emergency Department.

LGA: Large-for-gestational-age.

SGA: Small-for-gestational-age.

SVD: Spontaneous vaginal delivery.

the first day of the recorded birth month. This approximation may have introduced misclassification bias in age-specific analyses. For example, infants born later in the month were assigned an earlier estimated birthdate, which could have artificially inflated their calculated age. As a result, hospitalisations occurring at the end of the neonatal period (e.g., at 28 days of life) for these infants may have been misclassified into older age categories due to this inflated age estimate. This misclassification may have led to an underestimation of healthcare costs. Our cost estimation included only hospitalisation and ED presentations, excluding costs from GP or physician visits, which may lead to an underestimation of healthcare costs associated with adverse perinatal outcomes. We assumed that all births occurring in the state remained there and included them in our cost estimation. However, interstate and international migration may affect our estimation of hospitalisation, ED presentations and costs, probably leading to an underestimation [50]. The final limitation of our study, due to the unavailability of socioeconomic variable before 2013 in the Perinatal Trends dataset, is that we did not adjust for it in our model of cost drivers identification.

The stagnation of improvements in adverse perinatal outcomes such as PTB and LBW children in the NT over the past decades has placed a health burden and increased costs for the healthcare system through frequent hospital visits. The financial burden to the healthcare system is largely driven by extended hospital stays, vulnerability such as remote areas and Indigenous status, extreme of gestational age and birthweight, and also diagnosed and suspected congenital malformation. Thus, enhancing perinatal health interventions to reduce the length of birth hospitalisation and subsequent admissions or ED presentations may contribute to healthcare resource savings. However, such efforts should be balanced with ensuring safety and continuity of care, as shorter hospital stays may increase the risk of readmissions or adverse health outcomes if not carefully managed [51]. Furthermore, vulnerable populations, particularly children born to Indigenous parents and those living in remote areas, incur higher healthcare costs than the average. These findings highlight the

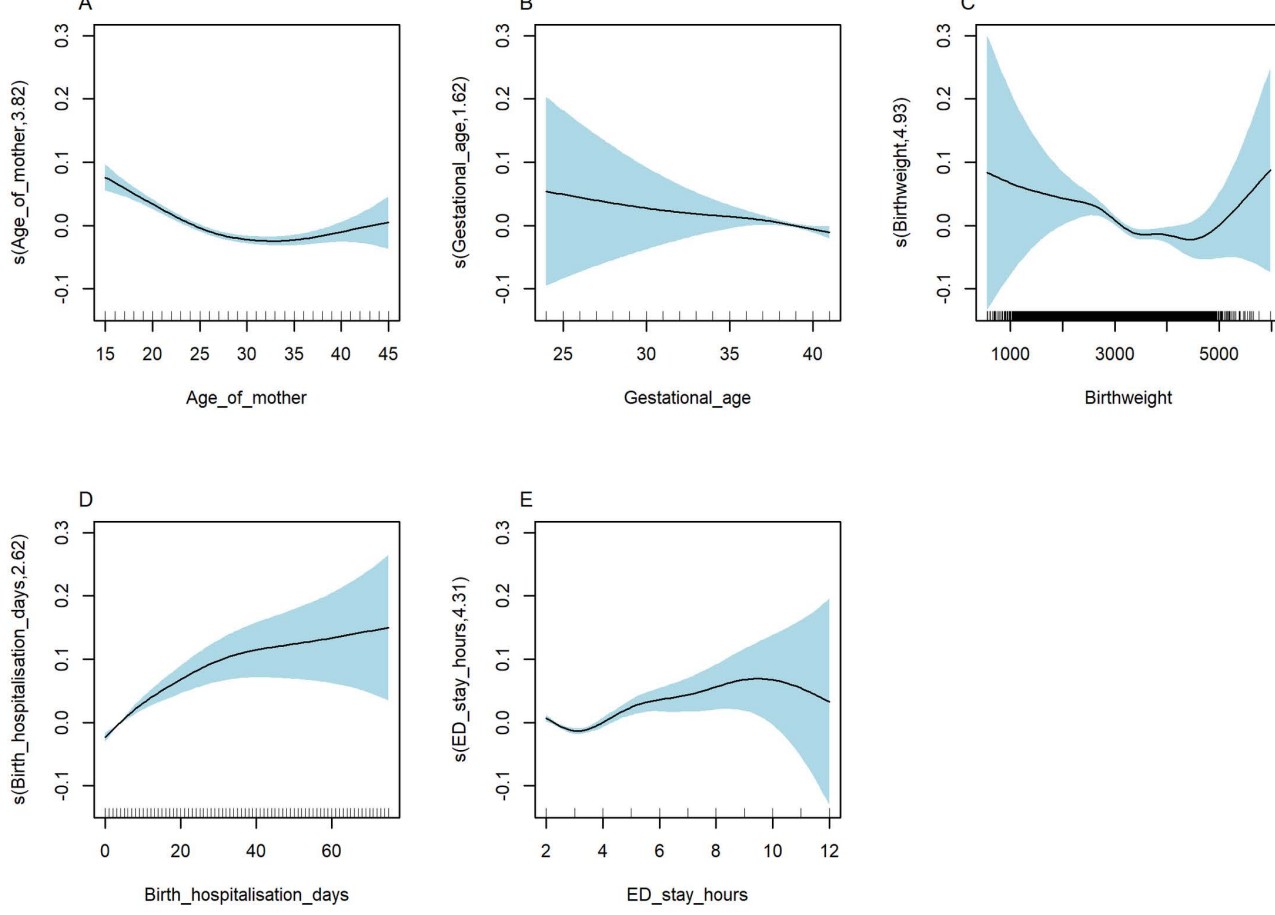

**Fig 9. Non-linear relationship of cost of ED presentations and covariates, NT, Australia, 2000–2020.** s(variable, df): Smooth function applied to the covariates, where df denotes the degree of freedom or smoothing parameter, indicating the flexibility of the non-linear relationship between the covariates and ED costs.

need for further research to investigate the association between length of stay and the risk of readmission among children with adverse perinatal outcomes. Additionally, quantifying the disparity of the burden across different population groups is essential to enable more targeted interventions aimed at reducing the associated dual health and economic impacts.

## Supporting information

**S1 Checklist. STROBE Statement—Checklist of items that should be included in reports of cohort studies.**
(DOCX)

**S1 Fig. Length of hospital stay by adverse perinatal outcomes from birth to age five years, NT, Australia, 2000 –2020;** A. Birth hospitalisation, B. Subsequent hospitalisation, and C. ED presentation.
(TIF)

**S2 Fig. Cost of hospitalisation by gestational age group and birthweight-for-gestational-age category, from birth to age five years, NT, Australia, 2000–2020.** PTB: Preterm birth
(TIF)

**S3 Fig. Cost of hospitalisation among pre(term) children by birthweight-for-gestational-age percentiles, NT, Australia, 2000–2020.** SAG-LBW: Small-for-gestational-age and low birthweight. AGA-LBW: Appropriate-for-gestational-age and low birthweight. AGA-Normal: Appropriate-for-gestational-age and normal birthweight. LGA-LBW: Large-for-gestational-age and low birthweight. LGA-Normal: Large-for-gestational-age and normal birthweight. LGA-Overwt: Large-for-gestational-age and overweight.
(TIF)

**S4 Fig. Interaction between gestational age and birthweight for hospitalisation costs from birth to age five years, NT, Australia, 2000–2020.**
(TIF)

**S5 Fig. Interaction between gestational age and birthweight for ED presentation costs from birth to age five years, NT, Australia, 2000–2020.**
(TIF)

**S1 Table. Cost of hospitalisation by adverse perinatal outcomes and years from birth to age five, NT, Australia, 2000–2020.**
(DOCX)

**S2 Table. Proportional mean cost by adverse perinatal outcome in NT, Australia, 2000–2020.**
(DOCX)

**S3 Table. Cost of hospitalisation by preterm categories from birth to age five years, NT, Australia, 2000–2020.**
(DOCX)

**S4 Table. Hospitalisation cost of PTB by different birthweight and gestational age for birthweight percentiles from birth to age five, NT, Australia, 2000–2020.**
(DOCX)

**S5 Table. Hospitalisation cost of term by different birthweight and gestational age for birthweight percentiles from birth to age five, NT, Australia, 2000–2020.**
(DOCX)

**S6 Table. Cost of ED presentation by adverse perinatal outcomes and years from birth to age five years, NT, Australia, 2000–2020.**
(DOCX)

**S7 Table. Cost of ED presentation by adverse perinatal outcomes and years from birth to age five years, NT, Australia, 2000–2020.**
(DOCX)

**S8 Table. Cost of ED presentations by preterm categories from birth to age five years, NT, Australia, 2000–2020.**
(DOCX)

**S9 Table. Drivers of hospitalisation cost from birth to age five years, NT, Australia, 2000–2020.**
(DOCX)

**S10 Table. Percentage change of non-linear co-variates and hospitalisation cost from birth to age five years, NT, Australia, 2000–2020.**
(DOCX)

**S11 Table. Drivers of cost of ED presentation from birth to age five years, NT, Australia, 2000–2020.**
(DOCX)

**S12 Table. Percentage change of non-linear co-variates and cost of ED presentations from birth to age five years, NT, Australia, 2000–2020.**
(DOCX)

**S13 Table. Drivers of cost of hospitalisation and ED presentation from birth to age five years, NT, Australia, 2000–2020.**
(DOCX)

## Acknowledgments

The authors would like to acknowledge the SA NT DataLink personnel for their technical and administrative assistance in linking the datasets and thank all data custodians for their support in data retrieval, preparation, and release.

## Author contributions

**Conceptualization:** Tsegaye G. Haile, Gavin Pereira, Richard Norman, Gizachew A. Tessema.

**Data curation:** Tsegaye G. Haile, Gizachew A. Tessema.

**Formal analysis:** Tsegaye G. Haile.

**Funding acquisition:** Gavin Pereira, Gizachew A. Tessema.

**Investigation:** Tsegaye G. Haile.

**Methodology:** Tsegaye G. Haile, Gavin Pereira, Richard Norman, Gizachew A. Tessema.

**Project administration:** Gavin Pereira, Richard Norman, Gizachew A. Tessema.

**Resources:** Gizachew A. Tessema.

**Software:** Tsegaye G. Haile.

**Supervision:** Gavin Pereira, Richard Norman, Gizachew A. Tessema.

**Validation:** Tsegaye G. Haile, Gavin Pereira, Richard Norman, Gizachew A. Tessema.

**Visualization:** Tsegaye G. Haile, Gavin Pereira, Richard Norman, Gizachew A. Tessema.

**Writing – original draft:** Tsegaye G. Haile.

**Writing – review & editing:** Tsegaye G. Haile, Gavin Pereira, Richard Norman, Gizachew A. Tessema.

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
