## [Decision Letter · Decision Letter 0]

5 Jun 2025

PGPH-D-25-00279

Trends in adverse perinatal outcomes and associated hospitalisations, emergency department presentations, and healthcare costs from birth to early childhood in Northern Territory, Australia: A two-decade population-based study

Dear Dr. Haile,

Thank you for submitting your manuscript to PLOS Global Public Health. After careful consideration, we feel that it has merit but does not fully meet PLOS Global Public Health’s publication criteria as it currently stands. Therefore, we invite you to submit a revised version of the manuscript that addresses the points raised during the review process.

We look forward to receiving your revised manuscript.

Kind regards,

Shiyam Sunder, MBBS, MSc epidemiology, PhD

Academic Editor

Journal Requirements:

Additional Editor Comments (if provided):

Reviewers' comments:

Reviewer's Responses to Questions

**Comments to the Author**

1. Does this manuscript meet PLOS Global Public Health’s publication criteria?

Reviewer #1: Yes

Reviewer #2: Yes

2. Has the statistical analysis been performed appropriately and rigorously?

Reviewer #1: Yes

Reviewer #2: Yes

3. Have the authors made all data underlying the findings in their manuscript fully available (please refer to the Data Availability Statement at the start of the manuscript PDF file)?

Reviewer #1: Yes

Reviewer #2: No

4. Is the manuscript presented in an intelligible fashion and written in standard English?

Reviewer #1: Yes

Reviewer #2: Yes

Reviewer #1: This manuscript addresses an important issue by examining the trends and healthcare burden of adverse perinatal outcomes. The authors should be commended for their use of a large, linked, population-based dataset and for employing advanced statistical methods to analyze cost drivers. Below are my specific few comments and suggestions.

1-The rationale for combining PTB, SGA, and LBW could be justified. These outcomes are interrelated but also distinct with different etiologies and implications.

2- The Eligibility criteria (Lines 129 to 134) lack clarity on congenital anomalies or neonatal death cases

3-Model selection methods are briefly mentioned (Lines 157 to 158), but detail on model diagnostics is not provided.

4-There is inconsistent reporting of missing data. Especially regarding values <6 (Line 245). How much data was affected?

5-Were interaction effects considered during analysis?

6-decreasing length of stay could lead to cost savings’ (Line 504) oversimplifies the issue. Shorter stays could increase readmissions or affect health outcomes also. This required rephrasing.

7-Also consider including the limitation of potential misclassification bias due to missing exact birthdates (Line 485)

Reviewer #2: Please see my comments below.

1. Page 1, abstract: it is stated "Mean hospitalizations increased for PTB (1.3 ± 0.7 to 6.9 ± 6.0), LBW (1.3 ± 0.6 to 7.2 ± 5.2), and SGA (1.2 ± 0.6 to 8.1 ± 15.1), while readmission durations decreased".

Are these measured in terms of days or weeks?

2. Page 7 author stated, "Examining the economic burden trends for children born with adverse perinatal outcomes throughout childhood and across populations can support efforts to enhance perinatal health interventions..."

What other populations were considered in the analysis?

3. Page 10. Please describe how missing data was handled in the study?

4. Page 16, table #2. Others is one of the categories of marital status.

Can author describe the category "others"?

5. Page 16, table #2. How parity is defined in the table.

6.Page 17, table#2. Mother's country of birth is categorized in Australia and others.

It would be interesting to see which countries; besides Australia, mothers were born in.

**Do you want your identity to be public for this peer review?** For information about this choice, including consent withdrawal, please see our Privacy Policy

Reviewer #1: No

Reviewer #2: No

---

## [Editor Report · Decision Letter 1]

10 Jul 2025

Trends in adverse perinatal outcomes and associated hospitalisations, emergency department presentations, and healthcare costs from birth to early childhood in the Northern Territory, Australia: A two-decade population-based study

PGPH-D-25-00279R1

Dear Mr. Haile,

We are pleased to inform you that your manuscript 'Trends in adverse perinatal outcomes and associated hospitalisations, emergency department presentations, and healthcare costs from birth to early childhood in the Northern Territory, Australia: A two-decade population-based study' has been provisionally accepted for publication in PLOS Global Public Health.

Best regards,

Shiyam Sunder, MBBS, MSc epidemiology, PhD

Academic Editor